# An Improved 2D Meshfree Radial Point Interpolation Method for Stress Concentration Evaluation of Welded Component

**Fuming Bao [1], Bingzhi Chen [2], Yanguang Zhao [1] and Xinglin Guo [1,*]**

[1]  Department of Engineering Mechanics, State Key Laboratory of Structural Analysis for Industrial Equipment, Dalian University of Technology, Dalian 116024, China; baofuming@dlut.edu.cn (F.B.); ygzhao81@dlut.edu.cn (Y.Z.)

[2]  School of Mechanical Engineering, Dalian Jiaotong University, Dalian 116028, China; b.z.chen@outlook.com

*  Correspondence: xlguo@dlut.edu.cn; Tel.: +86-189-4081-7891



**Featured Application: Our studies provide an approach to apply meshfree methods in consistently evaluating stress concentration of welded components.**

**Abstract:** The study of characterizing the stress concentration effects at welds is one of the most important research directions for predicting the fatigue life of welded components. Stress solutions at the weld toe obtained from conventional meshfree methods are strongly influenced by parameters used in the methods as a result of stress singularity. In this study, an improved 2D meshfree radial point interpolation method (RPIM) is proposed for stress concentration evaluation of a welded component. The stress solutions are insensitive to parameters used in the improved RPIM. The improved RPIM-based scheme for consistently calculating stress concentration factor (SCF) and stress intensity factor at weld toe are presented. Our studies provide a novel approach to apply global weak-form meshfree methods in consistently computing SCFs and stress intensity factors at welds.

**Keywords:** stress analysis; welded joint; meshfree radial point interpolation method; stress concentration

---

## 1. Introduction

Welded structures are widely used in engineering. The fatigue life of welded joints is of great concern for a number of years. Stress distributions at welds and fatigue behavior of welded joints are continually investigated [1–5]. Various codes and standards as well as recommended practices (e.g., [6,7]) are widely adopted for fatigue evaluation of welded joints. The fatigue life of a welded joint is significantly influenced by the stress concentration effect at welded joints [8–11]. The study of characterizing the stress concentration effects and achieving effective notch stress calculation at welds are important research directions in the field of predicting the fatigue life of welded components.

In recent years, meshfree methods are attracting more and more interests in computational mechanics for its advantages such as good self-adaptability. With the rapid development of meshfree techniques, various problems in science and engineering have been solved by meshfree methods [12–15]. Meshfree methods have been great potential topics and trends in computational mechanics. However, the stress solutions obtained by meshfree methods are often influenced by parameters used in these methods such as shape parameters, dimension of the influence domain and number of field nodes. Especially, the sensitivity of the stress solutions at notch location of weld toe increases significantly due to the presence of stress singularity at geometric discontinuity. As a result, the conventional meshfree methods are not realistic for effectively determining the notch stresses or stress concentration at the weld toe.

Fortunately, in order to solve the problem that the conventional finite element method (FEM)-based stress solutions at the weld toe were strongly influenced by the type and size of elements, Dong proposed a mesh-insensitive structural stress method (or named Battelle structural stress method) [16]. Based on the mesh-insensitive structural stress method, the stress concentration factor was consistently characterized and notch stress intensity factor at weld toe was effectively calculated [17]. The equivalent structural stress-based master S-N curve method was then developed to evaluate the fatigue life of weld components [8]. Taking advantage of the mesh-insensitive structural stress method, the mesh-insensitive structural strain method was then proposed extending the mesh-insensitive structural stress method to a low-cycle fatigue regime [18]. Since 2007, the mesh-insensitive structural stress method has been adopted as an alternative fatigue evaluation method for welded components by ASME Section VIII Division 2 Code [19]. In recent years, the theory and engineering application of the mesh-insensitive structural stress method has been carried out continuously [20–24].

The aim of this study is to propose an improved 2D meshfree radial point interpolation method (RPIM) whose stress solutions are insensitive to parameters such as shape parameters, dimension of the influence domain and number of field nodes by introducing the idea of above mesh-insensitive structural stress method. The essential idea of improved RPIM is that the stress solutions were obtained directly from field nodes forces to avoid using derivatives of shape functions. The novelty of our studies is to provide an approach to apply global weak-form meshfree method in consistently and effectively calculating SCFs and notch stress intensity factors at weld toe.

Early meshfree methods originated from the collocation methods in 1930s [25,26]. The well-known smooth particle hydrodynamics (SPH) method was proposed for solving problems in astrophysics [27,28]. In recent decades, various meshfree methods have been proposed, such as the element free Galerkin (EFG) method [29], meshfree radial point interpolation method (RPIM) [30,31], reproducing kernel particle method (RKPM) [32,33], material point method (MPM) [34,35], etc. According to the types of deriving equations, meshfree methods can be classified into three types: meshfree strong-form method, meshfree weak-form method and meshfree weak-strong (MWS) form method [36]. Compared with the strong-form methods, the mesh weak-form methods have some advantages in stability and calculation accuracy. In terms of weighted residual method, different meshfree weak form methods can be obtained by using different trial functions and weight functions. RPIM is a typical global weak-form meshfree method proposed by Liu and co-workers in 2001 [30,31]. Using radial basis functions (RBF) as trial functions, RPIM is robust for irregular nodal distributions. The RPIM has been successfully used to solve many problems [37–41]. In addition, some improved RPIMs and coupling technologies based on RPIM have emerged [42–45]. Unfortunately, the stress solutions at weld toe obtained by RPIM are strongly influenced by the parameters used in RPIM as a result of singularity at notch, which makes the conventional RPIM unable to effectively and consistently determine the stress concentration factor and stress intensity factors at the weld toe.

The rest of this paper is composed of the following items. A brief description of conventional RPIM for two-dimensional linear elasticity problem was presented. Then, the improved RPIM which modified the stress calculation process of conventional RPIM was developed for achieving stress calculation consistency at weld toe where existing singularity. Cantilever beam, a typical plate lap fillet welded joint and a T-butt welded joint were employed for numerical examples. The numerical example of cantilever beam was carried out to validate the consistency and accuracy of the improved RPIM based stress solutions at the general location of cantilever beam without concentration effect. The numerical example of plate lap fillet welded joint was to validate insensitivity and effectiveness of SCF at weld toe based on improved RPIM. The numerical example of T-butt welded joint was to validate the consistency and effectiveness of the stress intensity solution at weld toe based on improved RPIM.

## 2. An Improved RPIM for Stress Concentration Evaluation of Welded Component

In this section, the basic principle of RPIM for two-dimensional linear elasticity problem is presented before the improved RPIM for stress concentration evaluation of a welded component is exhibited.

### 2.1. Formulation of RPIM Shape Functions

In terms of RPIM, the approximation function $u(x)$ at a point of interest $x$ can be written as

$$u(x) = \sum_{i=1}^{n} R_i(x)a_i + \sum_{j=1}^{m} p_j(x)b_j = \mathbf{R}^T(x)\mathbf{a} + \mathbf{p}^T(x)\mathbf{b} \tag{1}$$

where $R_i(x)$ is a radial basis function, $p_j(x)$ is monomial in the space coordinates $x_T = [x, y]$, $n$ represents the number of radial basis function, $m$ represents the number of polynomial basis function. $a_i$ and $b_j$ are unknown constants. There exists variable $r$ in radial basis function $R_i(x)$:

$$r = \sqrt{(x - x_i)^2 + (y - y_i)^2} \tag{2}$$

The geometric significance of $r$ is the distance between the point at $x$ and node at $x_i$. With the variable $r$, the multi-quadrics (MQ) function can be used as the radial basis function (RBF):

$$R_i(x, y) = (r_i^2 + (a_c d_c)^2)^q \tag{3}$$

where $a_c$ and $q$ are shape parameters, $d_c$ is the characteristic length relating to the nodal spacing. Applying Equation (1) to all the $n$ nodes in the support domain of point $x$, we can obtain $n$ linear equations which can be express in the matrix form:

$$\mathbf{U_s} = \mathbf{R_0}\mathbf{a} + \mathbf{P_m}\mathbf{b} \tag{4}$$

where
$\mathbf{U_s}$ is the vector of nodal function values:

$$\mathbf{U_s} = \{\; u_1 \quad u_2 \quad \cdots \quad u_n\} \tag{5}$$

$\mathbf{a}$ is the vector of unknown coefficients:

$$\mathbf{a} = \begin{bmatrix} a_1 & a_2 & \cdots & a_n \end{bmatrix}^T \tag{6}$$

$\mathbf{b}$ is the vector of unknown coefficients:

$$\mathbf{b} = \begin{bmatrix} b_1 & b_2 & \cdots & b_m \end{bmatrix}^T \tag{7}$$

$\mathbf{R_0}$ is the moment matrix of RBFs:

$$\mathbf{R_0} = \begin{bmatrix} R_1(r_1) & R_2(r_1) & \cdots & R_n(r_1) \\ R_1(r_2) & R_2(r_2) & \cdots & R_n(r_2) \\ \vdots & \vdots & \cdots & \vdots \\ R_1(r_n) & R_2(r_n) & \cdots & R_n(r_n) \end{bmatrix}_{(n \times n)} \tag{8}$$

$P_m$ is the polynomial moment matrix:

$$P_m = \begin{bmatrix} 1 & x_1 & y_1 & \cdots & p_m(x_1) \\ 1 & x_2 & y_2 & \cdots & p_m(x_2) \\ \vdots & \vdots & \vdots & \ddots & \vdots \\ 1 & x_n & y_n & \cdots & p_m(x_n) \end{bmatrix}_{(n \times m)} \tag{9}$$

Additional $m$ equations are employed as constraint conditions:

$$\sum_{i=1}^{n} p_j(x_i) a_i = P_m^T a = 0, j = 1, 2 \cdots, m \tag{10}$$

Combing Equations (4) and (10) we get:

$$\widetilde{U}_s = \begin{bmatrix} U_s \\ 0 \end{bmatrix} = \begin{bmatrix} R_0 & P_m \\ P_m^T & 0 \end{bmatrix} \begin{Bmatrix} a \\ b \end{Bmatrix} = G a_0 \tag{11}$$

where

$$G = \begin{bmatrix} R_0 & P_m \\ P_m^T & 0 \end{bmatrix} \tag{12}$$

$$a_0 = \begin{Bmatrix} a \\ b \end{Bmatrix} \tag{13}$$

A unique solution can be obtained by solving Equation (11):

$$a_0 = \begin{Bmatrix} a \\ b \end{Bmatrix} = G^{-1} \widetilde{U}_s \tag{14}$$

Substituting Equation (14) back into Equation (1), we obtain:

$$u(x) = \begin{Bmatrix} R^T(x) & p^T(x) \end{Bmatrix} \begin{Bmatrix} a \\ b \end{Bmatrix} = \begin{Bmatrix} R^T(x) & p^T(x) \end{Bmatrix} G^{-1} \widetilde{U}_s = \widetilde{\Phi}^T(x) \widetilde{U}_s \tag{15}$$

Then, the approximation function $u(x)$ can be expressed as:

$$u(x) = \widetilde{\Phi}^T(x) \widetilde{U}_s = \begin{Bmatrix} \phi_1(x) & \phi_2(x) & \cdots & \phi_n(x) & \phi_{n+1}(x) & \cdots & \phi_{n+m}(x) \end{Bmatrix} \begin{Bmatrix} U_s \\ 0 \end{Bmatrix} \tag{16}$$

where the RPIM shape function corresponding to nodal displacements can be expressed as

$$\Phi^T(x) = \begin{Bmatrix} \phi_1(x) & \phi_2(x) & \cdots & \phi_n(x) \end{Bmatrix} \tag{17}$$

Finally, the approximation function $u(x)$ can be expressed as

$$u(x) = \Phi^T(x) U_s = \sum_{i=1}^{n} \phi_i u_i \tag{18}$$

*2.2. Discretization of Two-Dimensional Linear Elasticity Equations*

Consider the two-dimensional linear elasticity problem which is defined in the domain $\Omega$ with boundary $\Gamma$. The governing equations and boundary conditions are written as:

$$L^T \sigma + b = 0 \qquad \text{in} \quad \Omega \tag{19}$$

$$\sigma = D\varepsilon \qquad \text{in} \quad \Omega \tag{20}$$

$$\varepsilon = Lu \qquad \text{in} \quad \Omega \tag{21}$$

$$\sigma n = \bar{t} \qquad \text{on} \quad \Gamma_t \tag{22}$$

$$u = \bar{u} \qquad \text{on} \quad \Gamma_u \tag{23}$$

where $\bar{t}$ and $\bar{u}$ are prescribed values of traction and displacement on boundary, $n$ is vector of unit outward normal on boundary.

$L$ denotes the differential operator expressed as:

$$L = \begin{bmatrix} \partial/\partial x & 0 \\ 0 & \partial/\partial y \\ \partial/\partial y & \partial/\partial x \end{bmatrix} \tag{24}$$

$\sigma^T$ denotes the stress vector expressed as:

$$\sigma^T = \left\{ \begin{array}{ccc} \sigma_{xx} & \sigma_{yy} & \tau_{xy} \end{array} \right\} \tag{25}$$

$u^T$ denotes the displacement vector expressed as:

$$u^T = \left\{ \begin{array}{cc} u & v \end{array} \right\} \tag{26}$$

$b^T$ denotes the body force vector expressed as:

$$b^T = \left\{ \begin{array}{cc} b_x & b_y \end{array} \right\} \tag{27}$$

The weak form of equilibrium Equation (19) can be written in the form of:

$$\int_{\Omega} (L\delta u)^T (DLu) d\Omega - \int_{\Omega} \delta u^T b \, d\Omega - \int_{\Gamma_t} \delta u^T \bar{t} \, d\Gamma = 0 \tag{28}$$

Discretization of Equation (28) with Equation (18) yields

$$KU = F \tag{29}$$

where

$$K_{IJ} = \int_{\Omega} B_I^T D B_J d\Omega \tag{30}$$

$$F = F^{(b)} + F^{(t)} = \int_{\Omega} \Phi_I^T b \, d\Omega + \int_{\Gamma} \Phi_I^T \bar{t} \, d\Gamma \tag{31}$$

$$B_i = \begin{bmatrix} \phi_{i,x} & 0 \\ 0 & \phi_{i,y} \\ \phi_{i,y} & \phi_{i,x} \end{bmatrix} \tag{32}$$

$$D = \frac{E}{1-v^2} \begin{bmatrix} 1 & v & 0 \\ v & 1 & 0 \\ 0 & 0 & (1-v)/2 \end{bmatrix}, \; \textit{for plane stress problem}$$

$$D = \frac{E(1-v)}{(1+v)(1-2v)} \begin{bmatrix} 1 & v/(1-v) & 0 \\ v/(1-v) & 1 & 0 \\ 0 & 0 & (1-2v)/2(1-v) \end{bmatrix}, \; \textit{for plane strain problem} \tag{33}$$

$E$ is Young's modulus. $\nu$ is Poisson's ratio.

### 2.3. Numerical Integration

The Gauss numerical quadrature was used to calculate the global stiffness matrix $K$ in Equation (29) after the problem domain was discretized into background cells:

$$K = \sum_{k}^{n_c} \sum_{i=1}^{n_g} \underbrace{\widehat{w}_i B^T(x_{Qi}) DB(x_{Qi}) |J_{ik}^D|}_{K^{ik}} \tag{34}$$

where $n_c$ is the number of background cells, $n_g$ is the number of Gauss points used in each background cell, $\widehat{w}_i$ is the Gauss weighting factor for the ith Gauss point at $x_{Qi}$, $J_{ik}^D$ is the Jacobian matrix for the area integration of the background cell k at which the Gauss point $x_{Qi}$ is located.

### 2.4. Improved RPIM Based Stress Solutions for Stress Concentration Evaluation at Weld Toe

As shown in Figure 1, the through-thickness stress distribution at the weld toe can be decomposed into the equilibrium-equivalent stress part and a self-equilibrating notch stress part [8,19]. The equilibrium-equivalent stress part can be then decomposed into a membrane component (characterized by $\sigma_m$) and a bending component (characterized by $\sigma_b$). In present discussions, the transverse shear stress component is not considered. In fact, if the remote loading produces a significant shear force at section, the shear stress will have a greater effect on crack growth at the weld toe, a situation not considered in this paper. An improved RPIM is presented below that realizes the consistent stress solutions at and near weld toe.

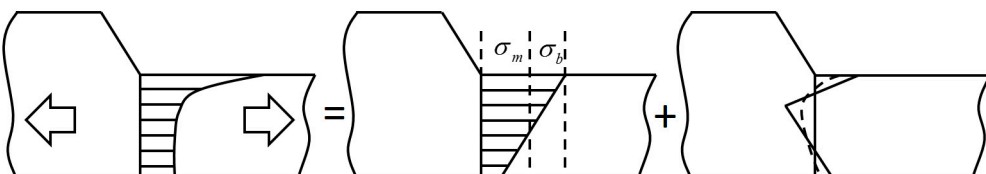

Part of equilibrium-equivalent stress    Part of self-equilibrating stress

**Figure 1.** Decomposition of through-thickness stress.

As shown in Figure 2, the section A-A at the location of interest and the reference section B-B with an arbitrary distance $\zeta$ from section A-A were introduced. Taking the continuum between A-A and B-B as the research object, the following equation must be satisfied according to the equilibrium conditions:

$$\sigma_m \cdot t = \int_{-t/2}^{t/2} \sigma_x(y) \cdot dy \tag{35}$$

$$\sigma_b \cdot \frac{t^2}{6} = \int_{-t/2}^{t/2} \sigma_x(y) \cdot y \cdot dy + \xi \cdot \int_{-t/2}^{t/2} \tau_{xy}(y) \cdot dy \tag{36}$$

where Equation (35) is the force balances equation in $x$ direction, and Equation (36) is the moment balances equation with respect to section A-A at $y = 0$. The physical significance of $\int_{-t/2}^{t/2} \sigma_x(y) \cdot dy$ is the internal force in the x direction on section B-B and the physical significance of $\int_{-t/2}^{t/2} \sigma_x(y) \cdot y \cdot dy$ is the internal moment on section B-B.

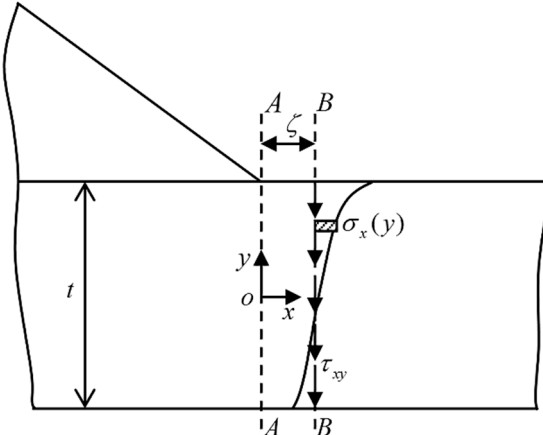

**Figure 2.** Through-thickness stress distributions in section B-B.

As the above integral terms can be interpreted as the internal force and internal moment between the right part and left part of continuum separated by section B-B, these integral terms can be obtained directly from the force of field nodes of the background cells through the following steps.

Step (1): Find all the field nodes within $\pm d_{ix}$ ($d_{ix}$ is the size of influence domain in the $x$ direction) from the section B-B. These field nodes were marked green as shown in Figure 3. The number of these field nodes is written as $n'$.

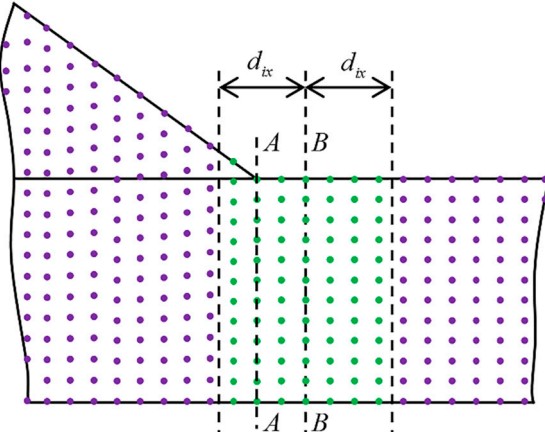

**Figure 3.** Illustration of field nodes within $\pm d_{ix}$ from the section B-B.

Step (2): In the right part of the continuum separated by section B-B, find all the background cells in which the support domain of Gauss points involves the field nodes mentioned in step (1). The number of these background cells is written as $n_{k'}$.

Step (3): The resultant force of the field nodes described in step (1) given by the background cells described in step (2) can be calculated by the following formula:

$$
\boldsymbol{F}^{internal} = \left\{
\begin{array}{c}
F_{1x}^{internal} \\
F_{1y}^{internal} \\
\vdots \\
F_{ix}^{internal} \\
F_{iy}^{internal} \\
\vdots \\
F_{nx}^{internal} \\
F_{ny}^{internal}
\end{array}
\right\} = \sum_{k'}^{n_{k'}} \sum_{i'=1}^{n_g} \widehat{w}_{i'} \boldsymbol{B}^T(x_{Qi'}) \boldsymbol{DB}(x_{Qi'}) \left| \boldsymbol{J}_{i'k'}^{D} \right| \cdot U \tag{37}
$$

where $x_{Qi'}$ represents the Gauss points described in step (2). $\boldsymbol{F}^{internal}$ is the $2n$ rows column vector representing the field node force in domain $\Omega$ given by the background cells described in step (2). Note that if field node $i$ is in the support domain of Gauss point of the background cell describe in step (2), the field node force $F_{ix}^{internal}$, $F_{iy}^{internal}$ denotes the field node force given by the background cells described in step (2). If field node $i$ is not in the support domain of the Gauss point of the background cell described in step (2), then:

$$F_{ix}^{internal} = 0 \tag{38}$$

$$F_{iy}^{internal} = 0 \tag{39}$$

which indicates that the background cells described in step (2) have no force effect on the field node $i$.

Step (4): The internal force in the x direction on section B-B can be expressed in the following form:

$$\int_{-t/2}^{t/2} \sigma_x(y) \cdot dy = \sum_{n'} F_{ix}^{internal} \tag{40}$$

The internal moment on section B-B can be expressed in the following form:

$$\int_{-t/2}^{t/2} \sigma_x(y) \cdot y \cdot dy = \sum_{n'} (F_{ix}^{internal} \cdot y_i) \tag{41}$$

Substituting Equations (40) and (41) back into Equations (35) and (36), we obtain

$$\sigma_m \cdot t = \sum_{n'} F_{ix}^{internal} \tag{42}$$

$$\sigma_b \cdot \frac{t^2}{6} = \sum_{n'} (F_{ix}^{internal} \cdot y_i) + \xi \cdot \int_{-t/2}^{t/2} \tau_{xy}(y) \cdot dy \tag{43}$$

Then, if $\xi \to 0$, we can obtain:

$$\sigma_m \cdot t = \sum_{n'} F_{ix}^{internal} \tag{44}$$

$$\sigma_b \cdot \frac{t^2}{6} = \sum_{n'} (F_{ix}^{internal} \cdot y_i) \tag{45}$$

The above flowchart for obtaining stress solutions in the form of membrane stress $\sigma_m$ and bending stress $\sigma_b$ was demonstrated Figure 4. Actually, the process demonstrated is also suitable for other global weak meshfree methods. The improved EFG, which modified the conventional EFG algorithm with the same process presented, and was also realized in our study. The results based on improved EFG were presented in Section 3. Limited by the length of the article, the basic principles of conventional EFG were not demonstrated in the paper. The detail of conventional EFG is given in [36].

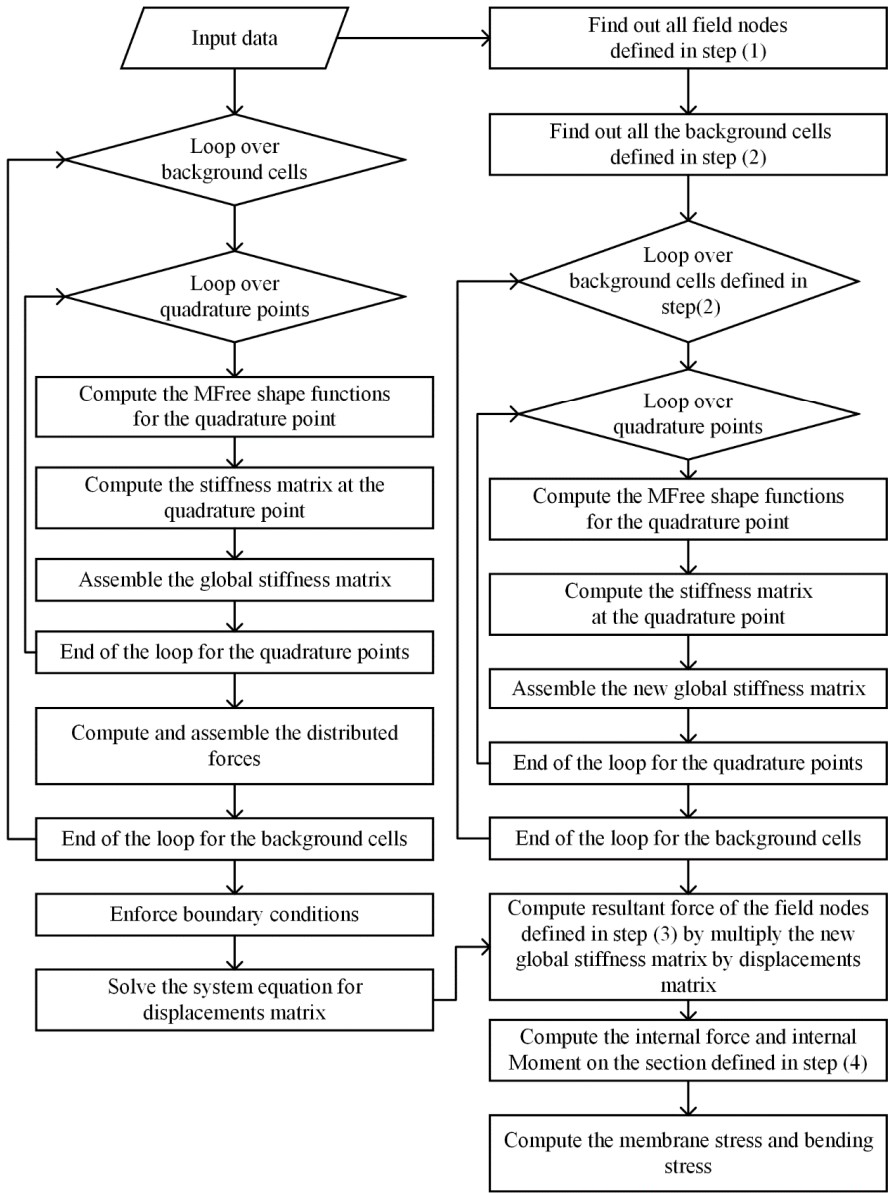

**Figure 4.** Flowchart of obtaining the stress solutions of improved radial point interpolation method (RPIM).

### 2.5. Notch Stress Intensity Factors Using Improved RPIM Based Stress Solutions

From the previous study [8], the notch stress effect at welds can be calculated by introducing characteristic depth $t_1$ ($t_1$ is taken as 0.1) in FEM model. The discussion on that the selection of $t_1$ should be selected as 0.1 was given in [8,17]. In this paper, the same value of $t_1$ ($t_1 = 0.1$) was introduced in meshfree model as shown in Figure 5a. The following Equation (46) must be satisfied by applying Equations (42) and (43) within region 1 and region 2, respectively (enforcing the equilibrium conditions and continuity condition within region 1 and region 2).

$$
\begin{aligned}
\sum_{region1+region2} F_{ix}^{internal} &= \sigma_1 t_1 + \tfrac{1}{2}(\sigma_2 - \sigma_1)t_1 + \sigma_2(t - t_1) + \tfrac{1}{2}(\sigma_3 - \sigma_2)(t - t_1) \\
\sum_{region1} (F_{ix}^{internal} \cdot y_i) &= (\sigma_1 t_1) \cdot \tfrac{t_1}{2} + \left[\tfrac{1}{2}(\sigma_1 - \sigma_2)t_1\right] \cdot \tfrac{2}{3} t_1 \\
\sum_{region2} (F_{ix}^{internal} \cdot y_i) &= \left[\sigma_3(t - t_1)\right] \cdot \tfrac{t - t_1}{2} + \left[\tfrac{1}{2}(\sigma_3 - \sigma_2)(t - t_1)\right] \cdot \tfrac{2}{3}(t - t_1)
\end{aligned}
\tag{46}
$$

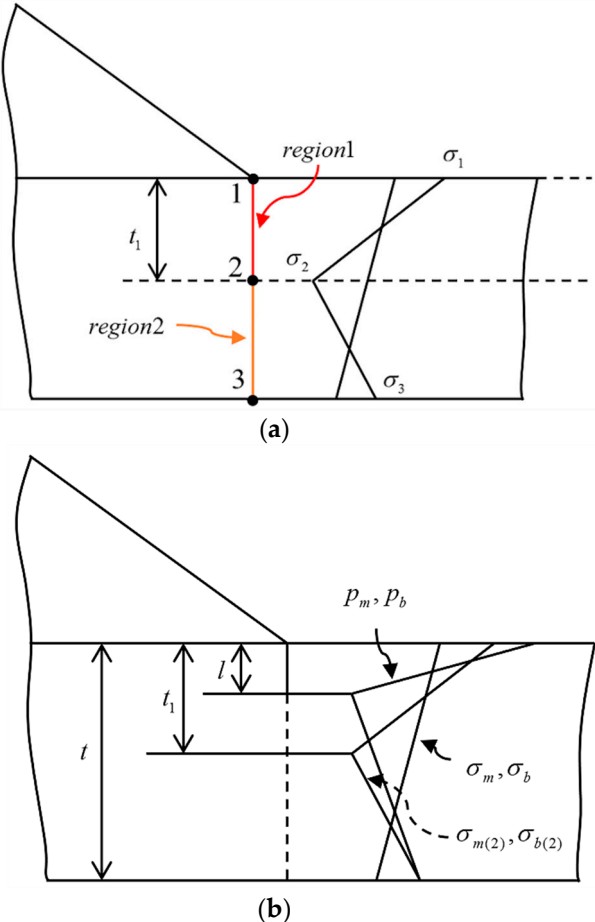

**Figure 5.** The distributions of notch stresses in terms of $p_m$, $p_m$ in Equation (48): (**a**) illustration of considering notch stress distribution as two linear distributions in region 1 and region 2; (**b**) relationship between $p_m$, $p_m$ in Equation (48) and $\sigma_{m(2)}, \sigma_{b(2)}, \sigma_m, \sigma_b$ in Equations (44), (45) and (47).

The membrane and bending components within region 1 and region 2 are as follows:

$$\begin{aligned} \sigma_{m(1)} &= \frac{\sigma_1+\sigma_2}{2}, \sigma_{b(1)} = \frac{\sigma_1-\sigma_2}{2} \\ \sigma_{m(1)} &= \frac{\sigma_2+\sigma_3}{2}, \sigma_{b(2)} = \frac{\sigma_2-\sigma_3}{2} \end{aligned} \tag{47}$$

With $\sigma_{m(2)}$, $\sigma_{b(2)}$ defined in Equation (47) and $\sigma_m$, $\sigma_b$ defined in Equations (44) and (45), the distributions of notch stresses can then be obtained in terms of $p_m$ and $p_b$ expressed as Equation (48) as shown in Figure 5b. The detailed discussion on deriving the relationship between $p_m$, $p_m$ and $\sigma_{m(2)}, \sigma_{b(2)}, \sigma_m, \sigma_b$ were given in [2].

$$\begin{aligned} p_m &= \left(\frac{t^2-3lt+2l^2}{2lt}\right)\sigma_{m(2)} + \left(\frac{t^2-3lt+2l^2}{2lt}\right)\sigma_{b(2)} - \left(\frac{t^2-5lt+2l^2}{2lt}\right)\sigma_m + \left(\frac{t-l}{2l}\right)\sigma_b \\ p_b &= \left(\frac{t^2+lt-2l^2}{2lt}\right)\sigma_{m(2)} + \left(\frac{t^2+lt-2l^2}{2lt}\right)\sigma_{b(2)} - \left(\frac{t^2+lt-2l^2}{2lt}\right)\sigma_m + \left(\frac{t+l}{2l}\right)\sigma_b \end{aligned} \tag{48}$$

Then, the stress intensity factor can be obtained based on $p_m, p_b$. For example, with applying superposition principles, the mode *I* stress intensity factor for edge cracks with notch effect is as follows:

$$K = p_s \sqrt{t}\left[f_m - r\frac{t}{a}(f_m - f_b)\right] \tag{49}$$

where $p_s = p_m + p_b$, $r = p_b/p_s$, $a = l$. $f_m(a/t)$ and $f_b(a/t)$ denotes the dimensionless functions documented in [17].

## 3. Numerical Examples

### 3.1. Cantilever Beam

Since the analytic solution for the cantilever problem is known, it is often employed as a benchmark for numerical methods. The sensitivity and accuracy of the improved RPIM based stress solutions for general cantilever beam problem (without stress concentration effect) were presented here. Consider a cantilever beam problem shown in Figure 6. Actually, because there is no stress concentration effect, the stress ($\sigma_m + \sigma_b$) here can be viewed as generalized nominal stresses.

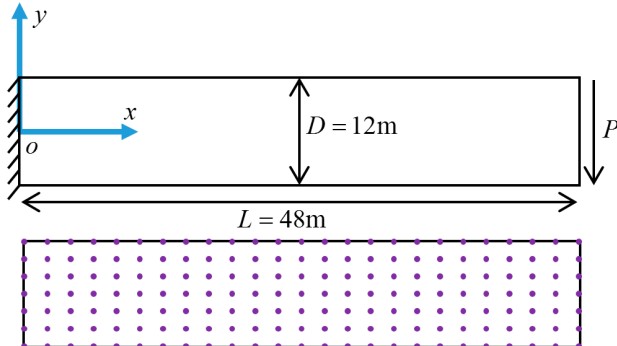

**Figure 6.** Model of cantilever beam and distribution of field nodes.

The thickness of the beam is 1 m. Young's modulus is 3E7. Poisson's ratio is 0.3. The model is considered as a plane stress problem. The parabolic distributed traction $P = 1000KN$ was enforced at the right free end as natural boundary condition. The analytic solution of this model was listed in [36]. A total of 175 (25 × 7) field nodes were distributed in the domain of the model. The domain was discretized into 40 (10 m × 4 m) rectangular background cells for Gauss numerical integrations. Penalty method was carried out to exert essential boundary conditions on the left end. The analytical solution, stress results of RPIM, EFG, improved RPIM and improved EFG at coordinate position (24 m, 6 m) were listed in Table 1. It can be seen that there are good agreements among the stress solutions of improved RPIM, improved EFG and the analytical results.

**Table 1.** The stress solutions obtained from different methods at coordinate position (24 m, 6 m).

| Method | Parameter | x (m) | y (m) | $\sigma_x$ (KPa) | Relative Error |
|---|---|---|---|---|---|
| Analytic Solution | - | | ±6.0 | ±1000.0 | - |
| RPIM | $a_c = 6, q = 1.03$ | 24.0 | ±6.0 | ±927.6 | 7.2% |
| Improved RPIM | $a_i = 3$ | | ±6.0 | ±1000.0 | 0.0 |
| Element Free Galerkin (EFG) | $a_i = 4$ | | ±6.0 | ±982.0 | 1.8% |
| Improved EFG | | | ±6.0 | ±1000.0 | 0.0 |

The performance of meshfree method is often affected by parameters used in the method. The stress solution sensitivity to parameters used in these methods was investigated below. In this study, the linear polynomial terms were used in the RPIM-MQ expressed as Equation (1). The linear basis and the cubic spline weight function were used in the moving least squares (MLS) approximation.

### 3.1.1. Sensitivity of Stress Solution to Shape Parameters

In conventional RPIM, the stress solutions are affected by shape function parameters in MQ-RBF. Wang and Liu reported that the RPIM performs the best when $q = 1.03$ or $q = 0.98$ [31]. The stress solutions obtained from conventional RPIM and improved RPIM for different $q$ ($q = -0.6$, 0.5, 1.03 and 1.25) with different $a_c$ ($a_c = 0.5$, 1.5, 3 and 5) were computed, respectively. Stress solutions at coordinate position (24 m, 6 m) are shown in Figure 7. Results show that the stress solutions obtained

from improved RPIM are insensitive to the shape parameters (both $q$ and $a_c$). The results also show that improved RPIM-based stress solutions are agreed with the analytical results.

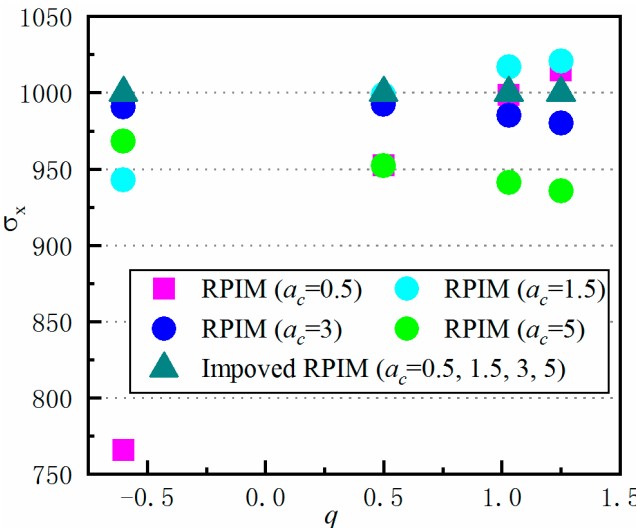

**Figure 7.** Investigation on sensitivity of stress solution to shape parameters.

### 3.1.2. Sensitivity of Stress Solutions to the Dimension of the Influence Domain

The size of rectangular influence domain can be defined simplistically by the following equations:

$$d_{ix} = a_i d_{cx}$$
$$d_{iy} = a_i d_{cy}$$
(50)

where $d_{cx}$ and $d_{cy}$ are the nodal spacing in the x and y directions, and $a_i$ is the dimensionless size of the influence domain. In conventional RPIM and EFG, the stress solutions are affected by the dimension of influence domain. Dimension of influence domain need to be carefully selected in order to get good stress result [36]. $d_{cx} = 2.4$ and $d_{cy} = 1.5$ were used in the numerical examples in this section. The stress solutions obtained from conventional RPIM, conventional EFG, improved RPIM and improved EFG with different dimension of influence domain ($a_i = 1.1, 1.5, 2, 2.5, 3,$ and $3.5$) were computed, respectively. Stress solutions at coordinate position (24 m, 6 m) are shown in Figure 8. Results show that the stress solutions obtained from improved RPIM and improved EFG are both insensitive to the dimension of influence domain. The results also show that there are good agreements among the stress solutions of improved RPIM, improved EFG and the analytical results.

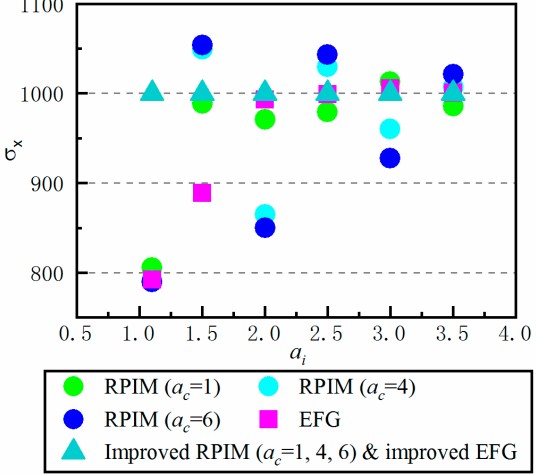

**Figure 8.** Investigation on sensitivity of stress solution to dimension of the influence domain.

### 3.1.3. Sensitivity of Stress Solution to Number of Field Nodes

Different numbers of regularly distributed field nodes (126, 175 and 403) were carried out, respectively, for numerical examinations to test the sensitivity of stress solution to the number of field nodes. Stress solutions at coordinate position (24 m, 6 m) were plotted in Figure 9. A model of regularly distributed 175 field nodes and 175 distributed irregularly field nodes was also carried out to test the sensitivity of stress solution to the distribution of field nodes. We found that the stress solution of based on improved RPIM and improved EFG were both insensitive to the field node distribution and the number of field nodes. The results also show that there are good agreements among the stress solution of improved RPIM, improved EFG and the analytical results.

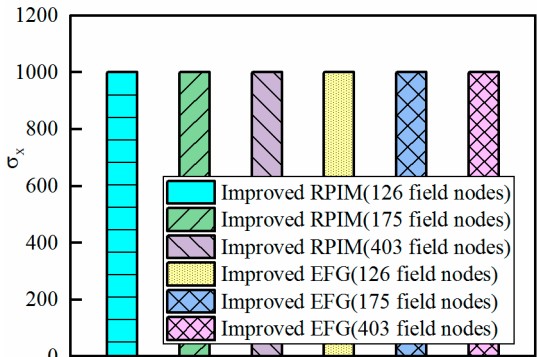

**Figure 9.** Investigation on sensitivity of stress solution to number of field nodes.

### 3.2. A Typical Plate Lap Fillet Weld

The typical plate lap fillet weld shown in Figure 10a was carried out to validate the consistency and effectiveness of SCF result at weld toe based on improved RPIM. The thickness of the plate lap fillet weld is 12 mm ($t = 12$ mm). The model is considered as plane strain model. The remote loading $F$ was enforced on the right end of this plate lap fillet weld. The values of structural stress ($\sigma_m + \sigma_b$) based SCFs using the stress solutions of improved RPIM can be written in the form of:

$$SCF = \frac{\sigma_m + \sigma_b}{F/A} \tag{51}$$

where $F$ denotes remote loading, and $A$ denotes the area of loaded member. As shown in Figure 10b, results show that the SCF based on the stress solutions of improved RPIM and improved EFG are both insensitive to shape parameters, dimension of the influence domain and number of field nodes. The SCF result of the same plate lap fillet based on mesh-insensitive structural stress method listed in [8] was also employed here as a validation for improved RPIM-based SCF. A good agreement between the result obtained from improved RPIM and results listed in [8] can be seen.

### 3.3. T-Butt Welded Joint

The consistency and effectiveness of the improved RPIM-based notch stress solution for T-butt welded joint are presented here. The T-butt welded joint is shown in Figure 11a. The thickness of the T-butt welded joint is 12 mm ($t = 12$ mm). The model is considered as plane strain model. The remote loading $F = 1200$ N was enforced on the right end of this T-butt welded joint. The detailed geometry conditions are shown in Figure 11a. The stress solution $\sigma_x$ at notch location of the T-butt welded joint obtained from conventional RPIM are presented in Figure 11b. As can be seen, stress solutions at weld toe obtained from conventional RPIM are sensitive to the parameters in RPIM, which makes it difficult to apply stress solution of the conventional RPIM in calculation SCFs or stress intensity factors. The SCFs based on the stress solutions of improved methods are plotted in Figure 11c. The mode I stress intensity factor for edge cracks obtained by Equation (49) was plotted in Figure 11d. The notch stress

intensities results of the same T-butt welded joint using mesh-insensitive structural stress method by Dong [17] and weight functions by Glinka citeB46-applsci-934788 were employed here as validations. It is noted that a small radius existed at notch root of weld in [46], and the radius was used for reducing singularity at welds. The corresponding notch stress intensities factor results are also shown in Figure 11d. The results show that there are good agreements among results based on solution of improved RPIM and results from [17,46].

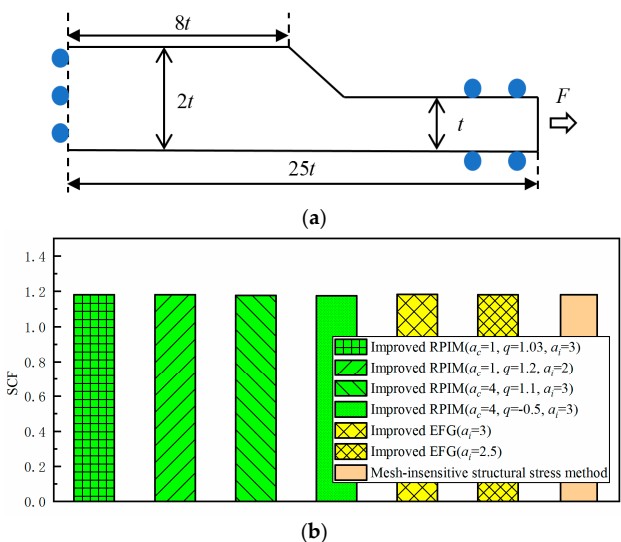

**Figure 10.** Comparisons of stress concentration factors (SCFs) obtained from different types of model: (**a**) model definition of the plate lap fillet weld (**b**) comparisons of SCFs based on improved RPIM, improved EFG and the mesh-insensitive structural stress method.

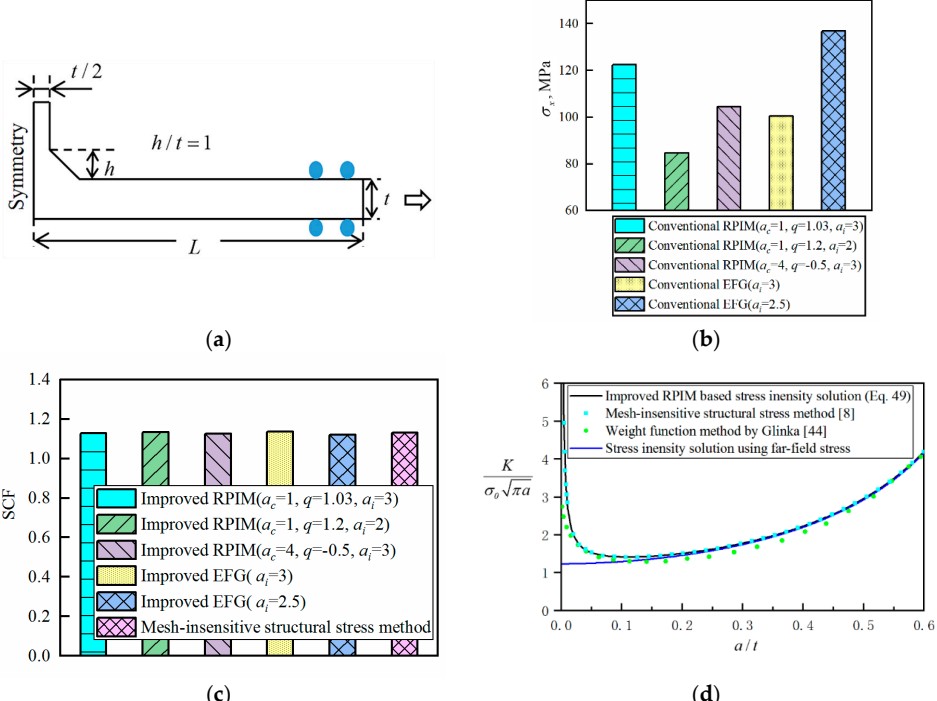

**Figure 11.** SCFs and notch stress intensities results of T-butt welded joint: (**a**) model definition; (**b**) stress solutions obtained from conventional RPIM; (**c**) results of SCFs based on solutions from improved RPIM and improved EFG; (**d**) validations of notch stress intensities results based on improved RPIM and improved EFG (the improved RPIM-based K was obtained from Equation (49)).

It can be seen from the first example that the stress solution of the conventional RPIM and EFG method are affected by the parameters used even if there is no stress concentration effect. The sensitivity of the stress solution increases obviously when there is existing singularity at the notch. The second and third examples demonstrated that the SCFs and stress intensity factors at the weld toe can be computed consistently and effectively based on stress solution of improved RPIM. Actually, the basic principle of the improved RPIM is that the discrete equations in the form of Equation (28) must satisfy mechanical equilibrium conditions regardless of shape parameters, dimension of the influence domain or number of field nodes. Since the equilibrium conditions are directly enforced on each field node, the most accurate solution variables in conventional RPIM are filed nodes forces. In improved RPIM, the stress solutions in the form of membrane stress and bending stress are obtained directly from field nodes forces in order to avoid using derivatives of shape functions. It is this reason that guarantees the insensitivity of the stress solutions of the improved RPIM.

## 4. Conclusions

Prior work has documented various meshfree methods used in science and engineering problems. However, when applying these meshfree methods in computing the SCF and stress intensity factor at notch of welds, the results are sensitive to the parameters used in these methods such as the existence of notch stress singularity. It is not realistic to calculate the stress concentration factor or stress intensity factor at welds by stress solution of the conventional RPIM.

In this study, an improved RPIM which modified the stress calculation process was presented to realize a process of obtaining consistent stress solutions at welds. The process of computing SCF and notch stress intensity factor at the weld toe based on the improved RPIM stress solution was presented. In the improved RPIM, the stress solutions in the form of membrane stress and bending stress were obtained directly from field nodes forces in order to avoid using derivatives of shape functions. Conclusions were summarized as follows.

(1) The generalized nominal stress of beam at general location without concentration effect can be computed consistently and effectively based on the stress solution of the improved RPIM

(2) The SCFs and stress intensity factors results at the weld toe can be computed consistently and effectively based on stress solution of the improved RPIM.

(3) Both the SCF and stress intensity factor results are insensitive to parameters such as shape parameters, dimension of the influence domain and number of field nodes.

(4) In addition, we also found that EFG with the same improved technique was also able to realize the process of obtaining consistent stress solutions at welds.

The main novelty of our studies is to provide an effective approach to apply meshfree methods in computing SCF and notch stress intensities at welds consistently. However, there are some limitations worth noting. Both RPIM and EFG are global weak-form meshfree methods. Further work should include how to apply local weak-form meshfree methods in consistently and effectively computing SCF and stress intensity factor at welds.

**Author Contributions:** Conceptualization, F.B. and B.C.; methodology, F.B.; validation, F.B., Y.Z. and X.G.; writing—original draft preparation, F.B.; All authors have read and agreed to the published version of the manuscript.

**Funding:** This research received no external funding.

**Acknowledgments:** We would like to thank the supports from the State Key Laboratory of Structural Analysis for Industrial Equipment of China.

**Conflicts of Interest:** The authors declare no conflict of interest.

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
