# Peer review of "An Improved 2D Meshfree Radial Point Interpolation Method for Stress Concentration Evaluation of Welded Component"

_applsci, doi:10.3390/app10196873_

Round 1

Reviewer 1 Report

Comments on the paper

  1. Check if all symbols and abbreviations have been explained, eg s1, a, ac, q, SCFs and others.
  2. There is no information about the program and FE used.
  3. What about residual stresses?
  4. What is the unit of the parameter K (Eq. 32) since in the equation p and f are dimensionless. The unit should be MPa m^1/2.
  5. There is a lot of ambiguity at work that needs to be clarified. There are no appropriate markings relating in the figures to the presented equations.
  6. It would be worthwhile to quote in the introduction also papers of: 1) ASTM E 739-80, Standard practice for statistical analysis of linearized stress-life (S-N) and strain-life (E-N) fatigue data, in: Annual Book of ASTM Standards, Vol.03.01. Philadelphia, 1989, pp. 667-673, 2) Rozumek D., Lewandowski J., Lesiuk G., Correia J., The influence of heat treatment on the behavior of fatigue crack growth in welded joints made of S355 under bending loading. Int. J. of Fatigue, Vol. 131, 2020.

Reviewer 2 Report

The topic of the manuscript entitled ‘An improved 2D meshfree radial point interpolation method for stress concentration evaluation of welded component’ falls within the scope of the journal Applied Sciences. The paper contains very interesting theoretical considerations and numerical results. It is of sufficient scientific interest and has originality in its technical content to merit publication. The authors have cited the relevant literature, but some interesting articles have been omitted.. Methods, interpretations of results and conclusions are correct and novel. The issues were well presented. In terms of content, the analysis does not raise any objections. The arrangement of work maintains substantive continuity and constitutes a logical whole. However, the manuscript is not suitable for publication in its present form.

The article requires small corrections.

Detailed comments are provided below.

Line 43: The year of publication may be omitted.

In line 59: the phrase 'In 1997' may be omitted.

Line 69: authors' initials should be omitted.

Line 135: the definition of the L matrix should be on a separate line as an equation and should be numbered. Likewise, the matrices on lines 144, 145 and 146.

Figure 2 and the caption under the figure should be on one page.

The equations on line 197 should be on separate lines and should be numbered.

The manuscript should be reorganized so that the bottom half of the 7th page is not blank.

Table 1 and its heading should be on one page.

Chapter Conclusions should be numbered. The most important conclusions should be presented in points.

Analysis of the current state of knowledge should take into account the following articles:

Winczek, J. A simplified method of predicting stresses in surfaced steel rods. J. Mater. Process. Technol. 2012, 212, pp. 1080 – 1088.

Fu, G.; Lourenco M.I.; Duan M.; Estefen S.F. Effect of boundary conditions on residual stress and distortion in T-joint welds. J. Constr. Steel Res. 2014, 102, pp. 121-135.

Moltasov, А.V. Stressed state of a butt-welded joint with regard for displacements of the centers of inertia. Mater. Sci. 2019, 55, pp. 358–366. doi:10.1007/s11003-019-00310-2.

Molski, K.L.; Tarasiuk, P. Stress concentration factors for butt-welded plates subjected to tensile, bending and shearing loads. Materials 2020,13, 1798; doi:10.3390/ma13081798.

Reviewer 3 Report

The paper addresses a very interesting and appealing research topic, the approach and the topics discussed in the paper are new and justify the interest in the publication. The content of the manuscript is good, and the presentation of information is quite clear. Some considerations could be evaluated before final acceptance:

  • Considering the current scientific literature, one aspect challenged by manuscript is related to the need for prediction of fatigue life of welded components (e.g. the conventional FEM). Following this observation, the authors stated: “The fatigue life of welded joint is significantly influenced by the stress concentration effect at welded joints”. In this respect, please extend the current literature review considering new studies investigating on this direction. In my experience, please let me suggest that the topics related to these approaches have been addressed by: 
    • C. Duan, S.Yang, J. Gu, Q. Xiong and Y.Wang. Study on Microstructure and Fatigue Damage Mechanism of 6082 Aluminum Alloy T-Type Metal Inert Gas (MIG) Welded Joint, Appl. Sci. 2018, 8(10);
    • G. Casalino, A.M. Losacco, A. Arnesano, F. Facchini, M. Pierangeli, C. Bonserio. Statistical Analysis and Modelling of an Yb: KGW Femtosecond Laser Micro-drilling, Process Procedia CIRP Volume 62, 2017, Pages 275-280.
  • Please detail the comparison of the results with regards to the stress solution and stress concentration factor (showed in fig. 9-10), obtained for different cases. The level of the parameters evaluated seems the same in all cases.  

Round 2

Reviewer 1 Report

The authors took into account all comments of the Reviewer. Based on the above mentioned, I recommend publishing the article in the journal Applied Sciences.

Author Response

Thank you very much for your careful review.